# Association between FT3 Levels and Exercise-Induced Cardiac Remodeling in Elite Athletes

**DOI:** 10.3390/biomedicines12071530

**Published:** 2024-07-10

**Authors:** Giuseppe Di Gioia, Maria Rosaria Squeo, Erika Lemme, Viviana Maestrini, Sara Monosilio, Armando Ferrera, Lorenzo Buzzelli, Daniele Valente, Antonio Pelliccia

**Affiliations:** 1Institute of Sports Medicine and Science, National Italian Olympic Committee, Largo Piero Gabrielli, 1, 00197 Rome, Italy; mariarosaria.squeo@coni.it (M.R.S.); erikalemme@msn.com (E.L.); viviana.maestrini@uniroma1.it (V.M.); sara.monosilio@uniroma1.it (S.M.); armando.ferrera@uniroma1.it (A.F.); ant.pelliccia@gmail.com (A.P.); 2Department of Movement, Human and Health Sciences, University of Rome “Foro Italico”, Piazza Lauro De Bosiis, 00135 Rome, Italy; 3Department of Clinical, Internal, Anesthesiology and Cardiovascular Sciences, Sapienza University of Rome, Viale del Policlinico, 155, 00161 Rome, Italy; 4Department of Clinical and Molecular Medicine, Sapienza University of Rome, 00189 Rome, Italy; 5Fondazione Policlinico Universitario Campus Bio-Medico, Via Álvaro del Portillo 200, 00128 Rome, Italy; lorenzo.buzzelli@unicampus.it (L.B.); daniele.valente@unicampus.it (D.V.)

**Keywords:** athletes, thyroid, heart, hormones, sport cardiology

## Abstract

Background: Previous studies demonstrated that variations of fT3, even within the euthyroid range, can influence cardiac function. Our aim was to investigate whether thyroid hormones, even within the euthyroid range, are associated with the magnitude of exercise-induced cardiac remodeling in Olympic athletes. Methods: We evaluated 1342 Olympic athletes (mean age 25.6 ± 5.1) practicing different sporting disciplines (power, skills, endurance, and mixed). Athletes underwent blood testing (thyroid stimulating hormone, fT3, and fT4), echocardiography, and exercise-stress testing. Athletes taking thyroid hormones, affected by thyroiditis, or presenting TSH out of ranges were excluded. Results: The level of thyroid hormones varied according to the type of sporting discipline practiced: endurance athletes presented the lowest TSH (*p* < 0.0001), fT3 (*p* = 0.007), and fT4 (*p* < 0.0001) in comparison to the remaining ones. Resting heart rate (HR) was positively correlated to fT3 in athletes of different disciplines (power: *p* = 0.0002, R2 = 0.04; skill: *p* = 0.0009, R2 = 0.05; endurance: *p* = 0.007, R2 = 0.03; and mixed: *p* = 0.04, R2 = 0.01). The same results were seen for peak HR in the exercise-stress test in athletes engaged in power, skill, and endurance (respectively, *p* < 0.0001, R2 = 0.04; *p* = 0.01, R2 = 0.04; and *p* = 0.005, R2 = 0.02). Moreover, a positive correlation was observed with cardiac dimensions, i.e., interventricular septum (power: *p* < 0.0001, R2 = 0.11; skill: *p* = 0.02, R2 = 0.03; endurance: *p* = 0.002, R2 = 0.03; mixed: *p* < 0.0001, R2 = 0.04). Furthermore, fT3 was directly correlated with the left ventricle (LV) end-diastolic volume in skills (*p* = 0.04, R2 = 0.03), endurance (*p* = 0.04, R2 = 0.01), and mixed (*p* = 0.04, R2 = 0.01). Conclusions: Thyroid hormones, even within the euthyroid range, are associated with cardiac adaptive response to exercise and may contribute to exercise-induced cardiac remodeling.

## 1. Introduction

Thyroid hormones, specifically free triiodothyronine (fT3), profoundly affect the heart and peripheral cardiovascular (CV) system, acting on cardiac myocytes and vascular smooth muscle [1,2,3,4,5,6,7,8]. The direct and indirect effects on CV hemodynamic result in increased cardiac inotropy and chronotropy, decreased systemic vascular resistance, and overall, increased cardiac output [1,2,3,4,5,6,7,8]. Furthermore, thyroid hormones can affect left ventricle (LV) structure through both increased hemodynamic load and the expression of structural and regulatory genes of the cardiac myocyte, in particular sarcoplasmic reticulum Ca2-ATPase (SERCa2) and myosin heavy chain (MHC) alpha levels. These mechanisms can lead to an increased protein synthesis in cardiac myocytes and eventually to thyroid hormone-induced myocardial hypertrophy [9,10,11,12,13].

Indeed, regardless of the well-characterized effects of subclinical or overt thyroid dysfunction on the heart, it has been hypothesized that variations in circulating thyroid hormone levels, even within the euthyroid range, could influence heart rate as well as cardiac structure and function [3,4,14,15]. The role of thyroid hormones in LV remodeling in clinically euthyroid subjects has been investigated only in a few studies [16,17,18]. Moreover, although the effects of training on the thyroid profile have been widely examined in the general population, no up-to-date data are available on the influence of thyroid function on cardiac morpho-functional parameters in athletes [19,20,21,22,23,24,25,26].

It is commonly known that intense and prolonged exercise induces adaptive cardiac changes, termed “athletes’ heart”, characterized by an increase in LV chamber size, wall thickness, and LV mass. [27,28,29,30,31,32]. Constitutional traits, along with the type of sport, are the major determinants (up to 60%) of the LV mass in athletes [33,34]. However, the potential role of thyroid hormones as modulators of athlete’s heart is unknown. Hence, we hypothesized that thyroid hormone levels may be one of the determining factors of morphological adaptation of the heart to physical training. Therefore, in the current study, we investigated the relationship between thyroid hormones and cardiac morpho-functional features in a large cohort of Olympic athletes.

## 2. Materials and Methods

We evaluated 1453 elite athletes who participated at Olympic and International competition in a 10-year period (from the London 2012 Summer Games to Beijing 2022 Winter Olympic Games). Athletes practiced several different sporting disciplines, arbitrarily divided into four categories, according to European Society of Cardiology classification and previous studies [35]:-(1) Power: weightlifting, Greco-Roman wrestling, judo, javelin, shot-putting, bobsleigh, skeleton, snowboard, swimming (under 800 m), alpine skiing, athletics (sprinting, shot putting, and discus), luge.-(2) Skill: archery, equestrian, golf, shooting, figure skating, sailing, curling, diving, equestrian sports.-(3) Endurance: cycling, rowing, canoeing, triathlon, long-distance running, long-distance swimming (over 800 m), cross-country skiing, pentathlon, biathlon, speed-skating, Nordic combined.-(4) Mixed: soccer, volleyball, basketball, tennis, fencing, water polo, rhythmic gymnastics, taekwondo, badminton, beach volley, softball.

Blood pressure was measured in the sitting position before exercise testing, as recommended [36,37]. A standard 12-lead ECG was performed in a supine position, and interpretation was fulfilled according to the international criteria for ECG interpretation in athletes [38]. All participants underwent maximal exercise testing on a bicycle ergometer (Cubestress XR400; Cardioline SpA, Milan, Italy) as previously reported, with an incremental protocol conducted until muscle exhaustion [39].

Body height and weight were obtained in each subject, and body mass index (BMI) was calculated as weight (kg)/height^2^ (m). Body surface area (BSA) was derived by the Mosteller formula [40].

Body composition and fat mass percentage were measured using Bioelectric Impedance Analysis (BIA 101 Quantum, Akern, Italy) using constant sinusoidal current at an intensity of 50 kHz and 400 μA.

Blood samples were collected in the early morning and after at least 10 h fasting and were analyzed on the same day. All blood tests (from 2012 to 2022) were collected and analyzed in the same laboratory. The following biochemical parameters were assessed: thyroid stimulating hormone (TSH), fT3, and free thyroxine (fT4) (with reference values for TSH between 0.27 and 4.2 uUI/mL; for fT3, between 3.1 and 6.8 pmol/L; and for fT4, between 12 and 22 pmol/L). Thyroid hormones were determined using immuno-electrochemiluminescence (Roche reagents) on Cobas 411 (Roche Diagnostics GmbH, Mannheim, Germany). We excluded athletes chronically taking thyroid hormone supplement, those with anamnestic thyroiditis, or those having TSH out of the normal range.

Two-dimensional echocardiography was performed with Philips EPIQ 7 (Philips Medical System, Andover, Massachusetts, USA). Cardiac images were obtained in multiple cross-sectional planes using standard transducer positions and then interpreted by two different expert cardiologists (GDG and AP). Measurements of end-diastolic and end-systolic LV cavity dimensions, interventricular septum (IVS), and posterior wall (LVPW) thickness were obtained according to the current recommendations [41,42]. LV mass (LVM) was calculated with the Devereux formula [43]. LV ejection fraction (LVEF) was calculated with the biplane method of disk summation, i.e., modified Simpson’s rule [43]. LV diastolic function was evaluated with pulsed-wave Doppler (PW) and tissue Doppler imaging, as recommended [44,45]. Inter-observer variability for LV chamber quantification was assessed in a sample of 100 athletes, selecting randomly one out of every ten athletes on our database, independently from sex and sporting discipline. Two investigators (GDG and AP), blinded, measured the same exam. Interclass correlation coefficients (ICC) was 0.91 (*p* < 0.0001) for inter-observer agreement.

The study design of the present investigation was evaluated and approved by the Review Board of the Institute of Medicine and Sports Science. All athletes included in this study were fully informed of the types and nature of the evaluation and signed the consent form according to Italian Law and Institute policy. All clinical data assembled from the study population are maintained in an institutional database.

### Statistical Analysis

Categorical variables were expressed as frequencies, and percentages were compared using Fisher’s exact test or Chi-square test as appropriate. Normality criteria were checked for any continuous variable, which were presented as mean ± standard deviation and compared using Student’s t-test for independent data if normally distributed. A *p*-value less than 0.05 was considered statistically significant. Dunn’s test and the Pairwise comparison method were used to focus on differences between sports and quartile comparison. The pooled *p*-values for the comparison test of four categories were determined, and if less than 0.05, a pairwise test was performed. Pairwise comparisons were considered significant if the *p*-value was less than 0.05. ICCs were calculated to assess inter-observer agreement. Statistical analysis was performed with STATA Statistics for Windows (SE, version 17).

## 3. Results

Our global study population included 1453 athletes (815 males, 56%; 57 Afro-Caribbean, 4%), mean age 25.6 ± 5.1 years old. Twenty-six athletes were chronically taking thyroid hormones due of clinical problems (19 were affected by hypothyroidism, 2 underwent total thyroidectomy, 4 suffered from Hashimoto’s thyroiditis, and 1 suffered from Basedow’s disease) In addition, 85 athletes showed TSH values out of the normal range: only 1 clinically manifested hypothyroidism (with elevated TSH and reduced ft4); 84 athletes were affected by sub-clinical hypothyroidism. All were excluded from the study.

Therefore, the remaining 1342 athletes (760 males, 56%) with normal TSH ranges were included in the present analysis. According to their sporting discipline, they were divided as follow: 29.8% power (400 athletes), 12.7% skill (171 athletes), 20.6% endurance (276 athletes), and 36.9% mixed sports (495 athletes). No relevant cardiovascular disease was found in our cohort.

In Table 1, the main differences in anthropometric, echocardiographic, exercise-stress test, and thyroid function are listed. All indicators were normally distributed.

As expected, males presented larger anthropometric parameters and significant cardiac morpho-functional differences compared to female counterparts. Moreover, differences in heart rate, blood pressure, and functional parameters were observed in the exercise-stress test. Also, thyroid function had sex-related peculiarity: although TSH values were similar (2.08 ± 0.8 uUI/mL in male vs. 2.03 ± 0.8 uUI/mL in female, *p* = 0.29), males compared to females showed higher ft3 (5.3 ± 0.6 pmol/L vs 4.7 ± 0.7 pmol/L, *p* < 0.0001) and ft4 values (16.7 ± 2.3 pmol/L vs. 15.4 ± 2.3 pmol/L, *p* < 0.0001).

Differences in thyroid hormones were also found according to the type of sporting disciplines practiced (Table 2).

In fact, endurance athletes presented lower TSH (1.93 ± 0.7 uUI/mL in endurance, 2 ± 0.8 uUI/mL in skill, 2.02 ± 0.8 uUI/mL in power, and 2.18 ± 0.8 uUI/mL in mixed; p pooled <0.0001, p pairwise: E vs. M, *p* <0.0001; P vs. M, *p* = 0.002; S vs. M, *p* = 0.009; remaining not significant), fT3 (4.9 ± 0.7 pmol/L in endurance, 5.1 ± 0.7 pmol/L in skill, 5.1 ± 0.7 pmol/L in power, and 5.1 ± 0.7 pmol/L in mixed; p pooled = 0.007; p pairwise: E vs. M, *p* = 0.001; P vs. E, *p* = 0.003; S vs. E, *p* = 0.023; remaining not significant), and fT4 (15.5 ± 2.2. pmol/L in endurance, 16.2 ± 2.2 pmol/L in skills, 16.5 ± 2.6 pmol/L in power, and 16.3 ± 2.3 pmol/L in mixed; p pooled < 0.0001; p pairwise P vs. E, *p* < 0.0001; E vs. M, *p* < 0.0001; S vs. E, *p* = 0.002; others not significant) values compared to other sporting disciplines. Moreover, hormone levels were different in male and female athletes; endurance males had the lowest TSH (*p* = 0.007), fT3 (*p* = 0.001), and fT4 levels (*p* < 0.0001). On the contrary, endurance females did not show significant differences in fT3 levels (*p* = 0.350), but lower TSH values (*p* = 0.016) in comparison to mixed disciplines (*p* = 0.004) and lower fT4 (*p* = 0.022), with significant differences compared to power sports (*p* = 0.015).

A linear relationship was found between the level of fT3 and certain cardiac morpho-functional parameters in all sporting disciplines. Namely, in all disciplines, resting heart rate (HR) was positively correlated to fT3 levels (power: *p* = 0.0002, R2 = 0.04; skills: *p* = 0.0009, R2 = 0.05; endurance: *p* = 0.007, R2 = 0.03; and mixed: *p* = 0.04, R2 = 0.01).

Also, peak HR at exercise-stress test was positively correlated to serum fT3 values in power, skill, and endurance disciplines (respectively, *p* < 0.0001, R2 = 0.04; *p* = 0.01, R2 = 0.04; and *p* = 0.005, R2 = 0.02).

Indeed, fT3 values were correlated with resting HR (Figure 1) both in males: *p* < 0.0001, R2 = 0.04 (Figure 1A) and females: *p* < 0.0001, R2 = 0.06 (Figure 1B) and with peak HR in the exercise-stress test (male: *p* = 0.0002, R2 = 0.02, Figure 1C; female: *p* < 0.0001, R2 = 0.06, Figure 1D).

Furthermore, fT3 values were positively correlated with ventricular septal thickness (Figure 2, Left Panel) independently of sporting discipline practiced (power: *p* < 0.0001, R2 = 0.11; skill: *p* = 0.02, R2 = 0.03; endurance: *p* = 0.002, R2 = 0.03; and mixed: *p* < 0.0001, R2 = 0.04) and with LV end-diastolic volume index (LVEDVi), as shown in Figure 2, Right Panel; they were directly correlated in skill (*p* = 0.04, R2 = 0.03), endurance (*p* = 0.04, R2 = 0.01), and mixed sports (*p* = 0.04, R2 = 0.01), while in power, they resulted close to statistical significance (*p* = 0.07, R2 = 0.009).

No significant correlations were observed between fT3 and LV function, as expressed by EF, in all sporting disciplines (power: *p* = 0.46, R2 = 0.00; skill: *p* = 0.20, R2 = 0.01; endurance: *p* = 0.51, R2 = 0.001; and mixed: *p* = 0.27, R2 = 0.003).

On the contrary, TSH and fT4 values were not correlated with HR, LVM indexed, LVEDV indexed, IVS, EF, E/A ratio, and E\E’.

As a further analysis, we arbitrarily divided the overall population according to fT3 quartiles (Table 3) to better assess the HR and IVS differences (Table 4). As shown in Figure 3, significant differences were observed, with HR increasing from quartile 1 to 4 in power (*p* = 0.026) and skill disciplines (*p* = 0.020), while in endurance and mixed disciplines, significant differences were found between quartile 1 and quartile 4 (*p* = 0.015 and *p* = 0.039, respectively). Also, IVS thickness (Figure 4) showed the same trend, increasing from quartile 1 to 4, with statistical significance in power and mixed disciplines (*p* < 0.0001 and *p* = 0.0002, respectively,), while in skill and endurance, significant differences were seen between quartile 1 and 4 (*p* = 0.019 and *p* = 0.013, respectively).

## 4. Discussion

Our study explored the relationship between thyroid function and cardiac morpho-functional parameters.

First, there is a positive association between fT3 levels and both resting and peak heart rate in the exercise test in most athletes regardless of the type of sport disciplines. This finding may likely be a consequence of the positive chronotropic effect of fT3 through binding to TR-alpha, the predominant thyroid receptor in myocardial tissue, and of its ability to modulate the β-adrenergic receptor expression [2,4,46]. Indeed, regarding morphologic cardiac features, a positive linear relationship exists between fT3 values and LV wall thickness, independently from sporting discipline practiced, along with the observation of a positive association with LV cavity dimensions in most of these athletes. The impact of thyroid activity is even better expressed by differences in both HR and LV wall thickness when athletic population was divided by fT3 quartiles, as shown in Table 3.

These data may suggest that small thyroid hormone variation, even within the euthyroid range, exerts concrete effects on the heart: cardiac exercise training adaptation may also be modulated by thyroid hormones, with an impact that has been largely underestimated in comparison to other well-described constitutional or sporting determinants [33]. Namely, thyroid function may affect the extent of LV hypertrophy and, consequently, ventricular mass. At the cellular level, the ability of thyroid hormones to induce myocardial hypertrophy has been explained by the activation of PI3K/AKT/mTOR and GK3β signaling pathways and an increased protein synthesis in the terminally differentiated cardiac myocyte, particularly sarcoplasmic reticulum Ca2-ATPase and myosin heavy chain [6,11,47].

This evidence has significant clinical implications, as the extent of myocardial adaptation is unique to each athlete, varying considerably among individuals, as a result of complex interactions between multiple factors [13,28,33,48,49]. Therefore, thyroid function, irrespective of the discipline practiced, should be taken into consideration in the differential diagnosis of physiologic LV remodeling from pathologic hypertrophy or to account for unexplained variations in heart rate.

A few previous studies conducted in the general, euthyroid, middle-aged population already confirmed the same relationship: Regarding HR, all these studies agree in describing a robust positive association of fT3 with cardiac frequency, in accord with our results in athletes [16,17,18]. Concerning cardiac structural parameters, Iida et al. [18] demonstrated that in 318 hypertensive patients without known thyroidal diseases, free peripheral thyroid hormones and TSH levels (even within the normal reference range) were associated with LVMi, the former positively and the latter inversely. In particular, fT3 was positively correlated with LV cavity size and wall thickness, suggesting an association for increasing fT3 values with greater LV hypertrophy. Similarly, Neves et al. [16] showed that in eight hundred thirty-five subjects aged ≥45 years, increasing fT3 had a non-linear, U-shaped association with larger LV cavity volumes and LV wall thickness, with a greater wall thickness near the extremes of the normal range of thyroid hormone concentrations. On the contrary, Roef et al. [17], in a population-representative random sample of patients aged between 35 and 55 years and free from overt cardiovascular disease at baseline, showed that fT3 was negatively associated with LVEDV, thus with a smaller LV cavity size and positively correlated with relative wall thickness, but the latter disappeared after adjustment for blood pressure. These data, albeit in different populations, support our results and confirm the potential role of thyroid hormones, particularly fT3 (even within normal limits), in myocardial remodeling.

On the other hand, we demonstrated that the type of sport discipline participated has also associated with thyroid hormone levels, with endurance athletes presenting the lowest TSH, fT3, and fT4, and that this effect appears to be greater in male athletes. These results are supported by previous studies [21,26,50], suggesting that chronic aerobic exercise could induce positive metabolic adaptation promoted by changes in thyroid balance, with the aim of improving athletic performance and response to long-term intense training. In fact, the status of energetic homeostasis is regulated by endocrine and central nervous systems, sensitive to external triggers, and in this scenario, chronic exercise represents an energy-saving signal to the human organism [24]. The candidate mechanisms whereby training exerts these effects can be attributed to exercise-induced negative energy balance as well as signaling alterations in the hypothalamus–pituitary–thyroid and hypothalamus–adipocyte–leptin axes [24]. Leptin, an adipocyte-secreted hormone with the signal function of energy availability, would act as a major endocrine modulator for the regulation of metabolic energy expenditure during exercise by altering hypothalamic neuropeptides [24,25,51,52]. The long-term training promotes a catabolic state with hormonal feedback from the depleted adipose tissue and induces decreased leptin levels. The latter in turn provides suppressed hypothalamic function and reduced central stimulation of TSH, resulting finally in reduced values of TSH, fT3, and fT4, as a possible energy-saving mechanism in exercising individuals [24,25,51,52]. Similarly, prolactin values have also been correlated with the thyroid response to intense and prolonged endurance exercise, suggesting an interaction between the two hormones in metabolic regulation following training [53,54].

### Limitations

Our study has several limitations. First, it is a retrospective cross-sectional study with all the consequent methodological limits of the data set in establishing an association without causation. However, sample size and sub-analysis by sporting discipline allow us to better understand the cause–effect relationship between exercise training influence on thyroid function and vice versa. Moreover, in our pre-event participation study, blood test screening in athletes includes a thyroid function evaluation though TSH, ft3, and fT4, but anti-thyroid autoantibodies (thyroid peroxidase antibodies, TPOAb) were not dosed. Therefore, in exclusion criteria, thyroiditis was identified only by anamnestic criteria. Finally, this was almost exclusively a Caucasian cohort that reduce generalizability to athletes from other ethnic backgrounds. 

## 5. Conclusions

Our study demonstrated a relationship between thyroid function and cardiovascular function in athletes. First, small variations in fT3 circulating levels, even within the normal range, are associated with rest and peak heart rate, the magnitude of LV wall thickness, end-diastolic volume, and mass in athletes from all sporting disciplines. Chronic aerobic training is also associated with lower thyroid hormone levels, possibly representing an adaptive mechanism.

Therefore, thyroid hormonal variation, even within the euthyroid range, may exert a relevant effect on the heart and cardiac remodeling. Further studies are necessary in order to conclude a causal effect between FT3 and TSH levels and exercise-induced cardiac remodeling in elite athletes.

## Figures and Tables

**Figure 1 biomedicines-12-01530-f001:**
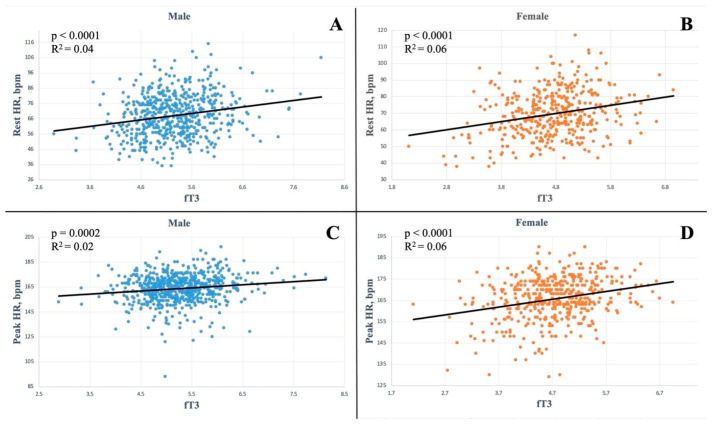
Linear regression analysis between gender and heart rate at rest and at peak of exercise-stress test. (**A**) Rest heart rate in male athletes; (**B**) rest heart rate in female athletes; (**C**) peak heart rate in male athletes; (**D**) peak heart rate in female athletes. Abbreviations: HR: heart rate.

**Figure 2 biomedicines-12-01530-f002:**
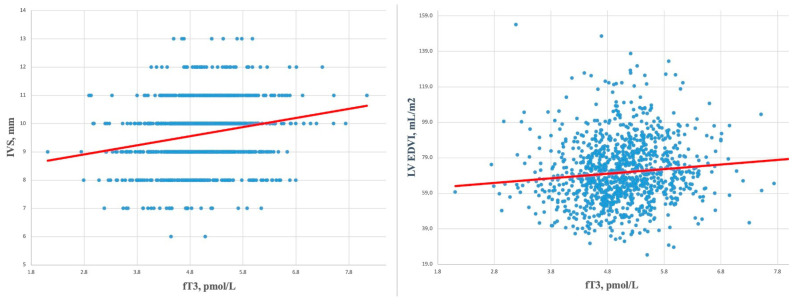
Linear regression analysis between serum fT3 values and interventricular septum thickness (**Left** Panel) and left ventricle end-diastolic volume indexed (**Right** Panel). Abbreviations: IVS: interventricular septum; LVEDVi: left ventricle end-diastolic volume indexed.

**Figure 3 biomedicines-12-01530-f003:**
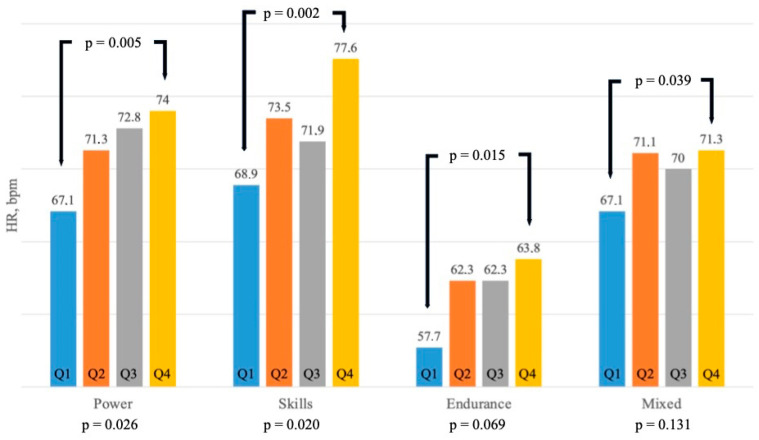
Variations in rest heart rate in different sporting disciplines according to fT3 quartiles. Abbreviations: HR: heart rate.

**Figure 4 biomedicines-12-01530-f004:**
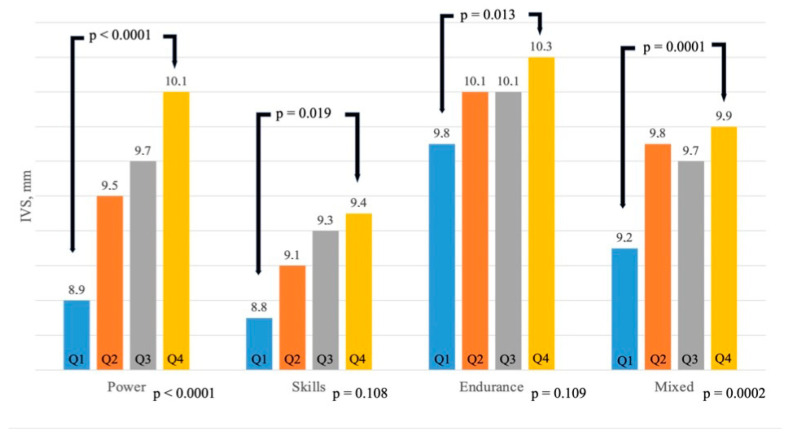
Variations in LV wall (interventricular septum) thickness in different sporting disciplines according to fT3 quartiles. Abbreviations: IVS: interventricular thickness.

**Table 1 biomedicines-12-01530-t001:** General/anthropometric data, echocardiographic values, exercise-stress test parameters, and thyroid function according to sex.

		Male	Female	*p*
	n, (%)	760 (56.6)	582 (43.4)	
Thyroid function	TSH, uUI/mL	2.08 ± 0.8	2.03 ± 0.8	0.29
	fT3, pmol/L	5.3 ± 0.6	4.7 ± 0.7	<0.0001
	fT4, pmol/L	16.7 ± 2.3	15.4 ± 2.3	<0.0001
General	Age, years	25.9 ± 5.2	25.4 ± 4.9	0.048
	Black, n (%)	32 (4.2)	24 (4.1)	0.99
	Smokers, n (%)	65 (8.5)	44 (7.6)	0.54
	BSA	2.02 ± 0.2	1.72 ± 0.2	<0.0001
	Weight, kg	81.2 ± 13.7	63.7 ± 10.6	<0.0001
	BMI, kg/m^2^	24.2 ± 3	22 ± 2.7	<0.0001
	Fat mass, %	11.8 ± 5.1	20.6 ± 5.3	<0.0001
Echocardiogram	LVEDDi, mm	27.3 ± 2.5	29 ± 3.9	<0.0001
	LVESDi, mm	16.9 ± 2.1	17.7 ± 2.8	<0.0001
	LVEDVi, mL	75.7 ± 17.6	63.2 ± 14.9	<0.0001
	LVESVi, mL	27.4 ± 7.6	22.3 ± 6.4	<0.0001
	IVS, mm	10.1 ± 1	8.9 ± 0.9	<0.0001
	PW, mm	9.8 ± 1	8.5 ± 0.4	<0.0001
	EF, %	63.8 ± 5	64.7 ± 5	0.0005
	LVMi, gr	105.2 ± 20.1	87.8 ± 18	<0.0001
	RWT	0.36 ± 0.03	0.34 ± 0.03	<0.0001
	LA, mm	37.1 ± 4.4	33.3 ± 3.6	<0.0001
	LAVi, mL	22.5 ± 7.1	19.7 ± 5.8	<0.0001
	AR, mm	32.3 ± 3.2	28 ± 2.8	<0.0001
	AA, mm	28 ± 3.1	25.4 ± 2.9	<0.0001
	PASP, mmHg	23.1 ± 4.3	22.2 ± 4	0.0003
	E wave, cm/s	82.1 ± 15.3	88.2 ± 15.4	<0.0001
	A wave, cm/s	45.7 ± 10	47.3 ± 11.6	0.006
	LV E’, m/s	12.4 ± 2.2	12.5 ± 2.2	0.41
	LV A’, m/s	6.8 ± 1.6	6.1 ± 1.3	<0.0001
	LV S’, m/s	8.3 ± 1.4	7.8 ± 1.3	<0.0001
	TAPSE, mm	27.2 ± 4.6	25.9 ± 3.9	<0.0001
	E/A	1.87 ± 0.5	1.96 ± 0.5	0.004
	E/E’	6.75 ± 1.4	7.21 ± 1.6	<0.0001
	RAi, mm^2^	9.9 ± 2.2	8.8 ± 2.1	0.0004
Stress-test	Rest HR, bpm	68.3 ± 13.5	69.5 ± 13.7	0.15
	Peak HR, bpm	164.1 ± 11.1	166.3 ± 10	0.0002
	MTHR, %	85 ± 5.4	85.7 ± 4.9	0.009
	Rest SBP, mmHg	114.5 ± 10.6	104.7 ± 10.4	<0.0001
	Rest DBP, mmHg	71.3 ± 8	65.7 ± 7.1	<0.0001
	Peak SBP, mmHg	185.2 ± 18.5	165.5 ± 15.7	<0.0001
	Peak DBP, mmHg	75.3 ± 8.6	71 ± 8.6	<0.0001
	Watt	279.4 ± 60.7	201.6 ± 41.6	<0.0001

Abbreviations: AA: ascending aorta; AR: aortic root; BMI: body mass index; BSA: body surface area; DBP: diastolic blood pressure; EF: ejection fraction; HR: heart rate; IVS: interventricular septum; LA: left atrium; LAVi: left atrial volume indexed; LV: left ventricle; LVEDDi: left ventricle end-diastolic diameter indexed; LVEDVi: left ventricle end-diastolic volume indexed; LVESDi: left ventricle end-systolic diameter indexed; LVESVi: left ventricle end-systolic volume indexed; LVMi: left ventricle mass indexed; MTHR: maximal theoretical heart rate; PASP: pulmonary artery systolic pressure; PW: posterior wall; RAi: right atrium indexed; RWT: relative wall thickness; SBP: systolic blood pressure; TAPSE: tricuspid annular plane systolic excursion; TSH: thyroid stimulating hormone.

**Table 2 biomedicines-12-01530-t002:** Circulating thyroid hormone level differences in different sporting disciplines according to sex.

		Power	Skill	Endurance	Mixed	P Pooled	P Pairwise
MALE	n, (%)	222 (29.2)	98 (12.9)	167 (22)	273 (35.9)		
	Age, years	25.6 ± 4.7	27.9 ± 6.1	26.6 ± 4.3	25.1 ± 5.5	<0.0001	P vs. S, *p* = 0.0003; P vs. E, *p* = 0.052; S vs. E, *p* = 0.035; S vs. M, *p* < 0.0001; E vs. M, *p* = 0.004; P vs. M, *p* = 0.257.
	TSH, uUI/mL	2.06 ± 0.8	1.98 ± 0.8	1.96 ± 0.8	2.20 ± 0.7	0.007	E vs. M, *p* = 0.001; P vs. M, *p* = 0.049; S vs. M, *p* = 0.019; P vs. S, *p* = 0.437; P vs. E, *p* = 0.192; S vs. E, *p* = 0.781
	fT3, pmol/L	5.3 ± 0.6	5.3 ± 0.7	5.1 ± 0.7	5.3 ± 0.7	0.001	E vs. M, *p* = 0.0004; P vs. E, *p* = 0.001; S vs. E, *p* = 0.010; P vs. S, *p* = 0.843; P vs. M, *p* = 0.649; S vs. M, *p* = 0.888
	fT4, pmol/L	17 ± 2.5	16.9 ± 2.2	15.8 ± 2.3	17.1 ± 2.1	<0.0001	E vs. M, *p* < 0.0001; P vs. E, *p* < 0.0001; S vs. E, *p* = 0.0006; P vs. S, *p* = 0.748; P vs. M, *p* = 0.538; S vs. M, *p* = 0.386
FEMALE	n, (%)	178 (30.6)	73 (12.5)	109 (18.7)	222 (38.1)		
	Age, years	24.7 ± 4.6	26.9 ± 6	26.4 ± 4.2	25 ± 4.6	0.0007	P vs. S, *p* = 0.002; P vs. E, *p* = 0.001; S vs. M, *p* = 0.048; E vs. M, *p* = 0.007; P vs. M, *p* = 0.573; S vs. E, *p* = 0.597
	TSH, uUI/mL	1.96 ± 0.8	2.01 ± 0.8	1.89 ± 0.7	2.16 ± 0.8	0.016	E vs. M, *p* = 0.004; P vs. M, *p* = 0.019; P vs. S, *p* = 0.672; P vs. E, *p* = 0.422; S vs. E, *p* = 0.278; S vs. M, *p* = 0.184
	fT3, pmol/L	4.8 ± 0.7	4.8 ± 0.6	4.6 ± 0.7	4.8 ± 0.7	0.350	P vs. S, *p* = 0.745; P vs. E, *p* = 0.100; P vs. M, *p* = 0.760; S vs. E, *p* = 0.287; S vs. M, *p* = 0.916; E vs. M, *p* = 0.140
	fT4, pmol/L	15.8 ± 2.6	15.4 ± 1.9	15.1 ± 2.1	15.1 ± 2.1	0.022	P vs. E, *p* = 0.015; P vs. M, *p* = 0.010; P vs. S, *p* = 0.190; S vs. E, *p* = 0.407; S vs. M, *p* = 0.509; E vs. M, *p* = 0.807

Abbreviations: TSH: thyroid stimulating hormone.

**Table 3 biomedicines-12-01530-t003:** Values of fT3 quartiles according to sporting disciplines.

	Q1	Q2	Q3	Q4
Power	3.12–4.68 (n = 89)	4.69–5.03 (n = 84)	5.04–5.50 (n = 88)	5.51–6.78 (n = 89)
Skill	3.23–4.59 (n = 36)	4.60–5.03 (n = 36)	5.04–5.53 (n = 34)	5.54–6.8 (n = 37)
Endurance	3.18–4.48 (n = 65)	4.49–4.94 (n = 66)	4.95–5.38 (n = 65)	5.39–6.8 (n = 64)
Mixed	3.11–4.64 (n = 98)	4.65–5.08 (n = 95)	5.09–5.52 (n = 96)	5.53–6.74 (n = 96)

**Table 4 biomedicines-12-01530-t004:** Variations in basal heart rate and interventricular septum thickness in different sporting disciplines according to fT3 quartiles.

	Q1	Q2	Q3	Q4	P Pooled	P Pairwise
Heart Rate, bpm						
Power	67.1 ± 11.3	71.3 ± 12.3	72.8 ± 14.5	74 ± 13.5	0.026	Q1 vs. Q4, *p* = 0.005
Skills	68.9 ± 10.5	73.5 ± 11.7	71.9 ± 11.6	77.6 ± 11.8	0.020	Q1 vs. Q4, *p* = 0.002
Endurance	57.7 ± 11.5	62.3 ± 13.1	62.3 ± 12.2	63.8 ± 13.9	0.069	Q1 vs. Q4, *p* = 0.015
Mixed	67.1 ± 13.8	71.1 ± 12.4	70 ± 13	71.3 ± 13	0.131	Q1 vs. Q4, *p* = 0.039
LV wall thickness (IVS), mm						
Power	8.94 ± 1	9.5 ± 1.2	9.7 ± 1	10.1 ± 1.2	<0.0001	Q1 vs. Q4, *p* < 0.0001
Skills	8.77 ± 1	9.13 ± 1.1	9.26 ± 0.9	9.4 ± 1.1	0.108	Q1 vs. Q4, *p* = 0.019
Endurance	9.8 ± 1.2	10.1 ± 1.1	10.1 ± 1.1	10.3 ± 0.9	0.109	Q1 vs. Q4, *p* = 0.013
Mixed	9.2 ± 1	9.8 ± 0.9	9.7 ± 1	9.9 ± 1.1	0.0002	Q1 vs. Q4, *p* = 0.0001

Abbreviations: IVS: interventricular. Septum; LV: left ventricle.

## Data Availability

De-identified participant data are available upon reasonable request from the corresponding author.

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
