# Peer review of "Association between FT3 Levels and Exercise-Induced Cardiac Remodeling in Elite Athletes"

_biomedicines, 2024, doi:10.3390/biomedicines12071530_

Round 1
Reviewer 1 Report
Comments and Suggestions for Authors
Thank you for the opportunity to review this article. The title of the article fully reflects the essence of the study. The topic has a certain relevance, since the results can explain the relationship between clinical and morphofunctional changes in the cardiovascular system of elite athletes depending on the function of the thyroid gland. I have some comments: 1. correlation analysis does not allow us to determine the contribution of changes in the level of free T3 to the development of morphofunctional disorders of the cardiovascular system; this requires regression analysis; 2. Table 1 provides a comparative analysis of various indicators between male and female athletes. It is not clear why the authors did this, since this difference is obvious. It is also unclear whether all indicators were normally distributed, since only average values ​​are presented. This should be indicated in the manuscript. 3. In Table 2, the share of male athletes is calculated incorrectly, since in total it is 228.4%. 4. Table 3 presents free T3 levels by quartile. It is unclear why individuals in the 4th quartile have free T3 levels significantly higher than the upper limit of normal levels of this hormone. Did these athletes have clinical signs of hyperthyroidism? If so, why were they not excluded from the study? What age group were they in since hormone levels vary with age? 5. The authors needlessly refused to present a conclusion, since it is in this part of the manuscript that it is necessary to substantiate the scientific and practical significance of the research results obtained. 6. Most references are outdated. They should be updated.
Reviewer 2 Report
Comments and Suggestions for Authors
The work is topical and of high scientific added value. Accurate and well-explained statistics justified the choice of the study cohort. It should be noted that the cohort includes high-class athletes with various workloads. It remained unclear what the authors meant by avoiding including a conclusion section in the article. A partial summary can be seen in the last paragraph of the discussion section [282 - 303]. If the reviewer could ask the authors for justifications, why this choice? There are no other objections. The article will undoubtedly arouse deep interest in the society of experts and interested parties and will certainly be cited in the following scientific articles. The practical significance of the study should also be noted.
Round 2
Reviewer 1 Report
Comments and Suggestions for Authors
Accept as it is